# Mosquitoes Larvicidal Activity of *Ocimum kilimandscharicum* Oil Formulation under Laboratory and Field-Simulated Conditions

**DOI:** 10.3390/insects13020203

**Published:** 2022-02-16

**Authors:** John Bwire Ochola, Clifford Maina Mutero, Rose Muthoni Marubu, Barbara Frei Haller, Ahmed Hassanali, Wilber Lwande

**Affiliations:** 1Bioprospecting Program, International Centre of Insect Physiology and Ecology, P.O. Box 30772, Nairobi 00100, Kenya; cmutero@icipe.org (C.M.M.); rmarubu@icipe.org (R.M.M.); bfreihaller@bluewin.ch (B.F.H.); wlwande@yahoo.com (W.L.); 2Chemistry Department, School of Pure and Applied Sciences, Kenyatta University, P.O. Box 43844, Nairobi 00100, Kenya; ahmedhassanali786@gmail.com; 3Institute for Sustainable Malaria Control, School of Health Systems and Public Health, University of Pretoria, Private Bag X323, Pretoria 0001, South Africa; 4Institute of Pharmaceutical Sciences, ETH Zurich Vladimir-Prelog-Weg 1-5/10, 8093 Zurich, Switzerland

**Keywords:** larvicide, malaria, mosquito control, botanical, formulation, biopesticide

## Abstract

**Simple Summary:**

Mosquitoes are vectors of many severe diseases, notably malaria, yellow as well as dengue fever, and lymphatic filariasis. Vector control with synthetic chemical insecticides has been associated with resistance development and undesirable human and ecological effects. *Ocimum kilimandscharicum* oil formulation was evaluated for larvicidal activity against third instar mosquito larvae in the laboratory. The formulation was then compared with *Bacillus thuringiensis* subsp. *israelensis (Bti*) granules on *An. gambiae* larvae under field-simulated field trials. The LC_50_ for *O. kilimandscharicum* oil after 24 h against third instar larvae of *An. gambiae* was 0.74 ppm while for the emulsified *O. kilimandscharicum* oil formulation against third instar larvae of *An. gambiae* and *An. arabiensis* was 0.07 and 0.31 ppm, respectively. The high bioactivity and sublethal toxic effects to offspring of treated mosquito larvae in terms of the disruption of larval morphological aspects suggest its high potential as a botanical larvicide for the control of disease vectors. The bioactive formulation had the advantage of high solubility in aqueous media; it is also easily produced, ecofriendly, and low-cost. Moreover, because *O. kilimandscharicum* can easily be widely cultivated and has high EO yields, it may provide a valuable alternative for the effective and eco-friendly control of disease vectors among developing and developed communities.

**Abstract:**

Mosquitoes are vectors of many severe diseases, including malaria, yellow as well as dengue fever, and lymphatic filariasis. The use of synthetic chemical insecticides for mosquito control has been associated with resistance development and detrimental human, and ecological effects. For a safer alternative, the emulsified *Ocimum kilimandscharicum* oil formulation was evaluated for its larvicidal activity. The oil was analyzed by GC and GC/MS. The formulations were evaluated against third instar mosquito larvae in the laboratory and later compared with *Bacillus thuringiensis* subsp. *israelensis* against *An. gambiae* under field-simulated conditions. Thirty-nine compounds were identified in the oil, the main ones being D-camphor (36.6%) and limonene (18.6%). The formulation showed significant larval mortalities against *An. gambiae* and *An. arabiensis* larvae with LC_50_ of 0.07 and 0.31 ppm, respectively, at 24 h. Under the field-simulated trial, within 24 h, the formulation showed 98% mortality while *Bti* had achieved 54%. On day three, it caused 100% mortality while *Bti* achieved 76.5%. The high bioactivity and sublethal toxic effects to offspring of treated mosquito larvae, in terms of disruption of larval morphological aspects, suggest the high potential of the formulation as a botanical larvicide. The formulation, thus, may provide a valuable alternative for the effective and eco-friendly control of disease vectors.

## 1. Introduction

Mosquitoes act as vectors of numerous harmful human diseases prevalent in over 100 tropical, subtropical, and other countries of the world [1,2]. These include malaria, yellow fever, dengue fever, chikungunya, filariasis, West Nile virus, and Japanese encephalitis, which all lead to massive morbidity and mortality and are a major economic burden within disease-endemic countries [3]. For many years, synthetic insecticides, including pyrethroids, organochlorine, organophosphate, and carbamate compounds, have been used as the main tools for mosquito control. However, these have had major disadvantages related to their harmful effects to the environment, humans, and other non-target organisms, as well as the loss of efficacy after repeated usage resulting from resistance to the insecticides [4,5,6,7]. As a result, vector-borne diseases appear to be re-emerging, as observed in the last two decades [8]. This has prompted the need for the identification of new eco-friendly and sustainable alternatives for their control. Such methods involve environmental management [9] and, more importantly, the use of botanicals that are eco-friendly, easily accessible at low cost, and safe for human health and the environment [10,11,12,13]. Unlike conventional insecticides, which are based on single active ingredients, plant-derived products constitute blends of compounds that act additively and synergistically on the behavior and/or physiological processes of insects. Moreover, natural phytochemical blends show resistance-mitigating effects over long periods [14].

Defense against insect vectors and mosquitoes is based on three essential modes of approach. Firstly, a reduction in the population density of adults by killing them. However, there have been shifting behaviors of target mosquitoes, such as changes in the locations of blood-feeding from indoor to outdoor [15]. The second strategy involves preventing insects from sucking blood from the host using repellents. Thirdly, and more importantly, concerns larval source management (LSM), which utilizes insecticides or larvicides to kill mosquito larvae [16,17]. Mosquito larvae, unlike adults, cannot shift from their habitat to avoid control activities, making LSM a more convenient strategy [17,18,19]. Thus, the focus on decreasing the mosquito population at the larval stage has the benefit of managing the vector before acquisition and spreading of the disease and disrupting its life cycle. Among the LSM strategic approaches, the usage of insect growth inhibitors, such as methoprene; organophosphate insecticides, such as temephos; and bacterial insecticides, such as *Bt* subsp. *israelensis* (*Bti*), is now extensively utilized [20,21,22,23,24]. However, these methods are not economically accessible to most rural communities and there have been special concerns on the longer-term use of synthetic chemicals in the environment and the potential of mosquito larvae to develop resistance to the chemical control tools [23].

In recent studies, great efforts have been directed to aromatic plants that produce essential oils (EOs) which contain substances with insecticidal effects. These studies explore the efficacy of certain aspects of botanical larvicides as well as their practical application against vectors in relation to public health [23,24,25,26,27]. Components contained in EOs not only display various mechanisms of action but also numerous levels of ability to infiltrate the insect cuticle and penetrate their bodies, which is precisely associated with the capability to deliver an insecticidal impact. The most efficient EOs evaluated as larvicides include *Blumea densiflora*, *Auxemma glazioviana*, *Callitris glaucophylla*, *Cinnamomum microphyllum*, *Cinnamomum mollissimum*, *Cinnamomum rhyncophyllum*, and *Zanthoxylum oxyphyllum*, which gave LC_50_ < 10 ppm. Their chemical composition is mainly from the group of sesquiterpenes, aromatic acids, and ketones [26,28]. However, most of the previous studies on essential oils for use as larvicides failed to consider their low solubility in aqueous media. The low solubility results in very limited physiological effects on the larvae. Nevertheless, even at low doses in the aqueous media, oil droplets move up and coalesce and can form thin layers at the surface with reduced surface tension. This can make it extremely challenging for mosquito larvae to efficiently hang on the surface and inhale. As such, the detected adverse effects on the larvae can be caused largely by their failure to breathe on the surface during the period of exposure. This would, by extension, affect the nontarget aquatic organisms [8]. Moreover, the oil on the surface would also evaporate, thus rapidly limiting the duration of its negative effects on the larvae [27,28,29,30,31]. However, the low persistence of the effect of EOs and the formation of the film can be resolved through suitable formulation, such as using some encapsulation methods or oil in water nano-emulsions [31].

One of the most studied botanical sources is the *Ocimum kilimandscharicum* Gürke, whose extracts have shown considerable lethal activity against insect pests of agricultural and public health importance [32]. *O. kilimandscharicum* is a plant of the family Lamiaceae [33]. It is an important aromatic perennial evergreen shrub native to East Africa [34,35,36]. Its pharmacological potential or medicinal benefits have been well reported in the literature [35,36,37], but relatively few studies have been carried out on its larvicidal property against mosquitoes and formulation for its practical downstream application. The plant contains a rich reservoir of chemical constituents [38], some of which have known insecticidal properties. Therefore, this study outlines the results of the evaluation of the larvicidal potential of *O. kilimandscharicum* essential oil against *An. gambiae*, presents the use of a low-energy nano-emulsion delivery system with its volatile oil, and evaluates its larvicidal activity against two mosquito species under laboratory conditions. In addition, the performance of the water-based formulation against *An. gambiae* in field-simulated conditions was assessed. Furthermore, the effect of sublethal doses to post-larval stages in *An.gambiae* and *An. arabiensis* by the emulsified *O. kilimandscharicum* oil formulation was examined.

## 2. Materials and Methods

### 2.1. Plant Collection

The aerial parts of *O. kilimandscharicum* were collected from Isecheno Village, Kakamega County in western Kenya (0°17′ N; 34°45′ E). The environmental attributes, including location coordinates, were taken by a global positioning system (GPS) device, and a brief description of the plant sample was recorded in a notebook. The sample specimens were identified by a plant Taxonomist, Mr. Simon Mathenge (posthumous). The voucher specimens and OK/KAK/01/05 and OK/KAK/02/10 of the plants were deposited at the Herbarium of the National Museums of Kenya. The collected plant materials (leaves, flowers, or whole aerial part) were dried under shade at room temperature (25 ± 2 °C) for one week.

### 2.2. Extraction of Essential Oil of O. kilimandscharicum

Volatile essential oil from dried plant materials of *O. kilimandscharicum* was obtained by hydro-distillation with a Clevenger apparatus. Various quantities of each of the plant material were put into a ten-litre round-bottom flask, and water was added in the ratio of 1:3 (1 part of plant sample to 3 parts of water). The flask was then fitted with the Clevenger apparatus and a double pocket condenser. The plant materials were hydro-distilled for 4 h. The extracted essential oil was collected on the water layer in the Clevenger apparatus. The procedure was repeated three times for each plant sample. The organic and aqueous layers were then left to separate in a separating funnel. The hexane extract was then dried with anhydrous sodium sulphate to remove traces of the water and then filtered. The solvent was then removed by distillation under reduced pressure. The yield of essential oil was determined, and the oil was stored in amber-colored vials at 4 °C until used.

### 2.3. Identification of Constituents of O. kilimandscharicum Essential Oil

The *O. kilimandscharicum* oils were analyzed using a gas chromatograph (GC) (HP-7890A, Agilent Technologies, Wilmington, NC, USA) linked to a mass spectrometer (MS) operated in the electron impact mode (HP 5975 C, Agilent, Wilmington, NC, USA). The apparatus was equipped with a non-polar HP-5MS capillary column (30 m × 0.25 mm i.d.; 0.25-μm film thickness, with 5% phenylmethyl silicone as the stationary phase; J & W Scientific, Folsom, CA, USA). Helium (1.2 mL min^−1^) was used as the carrier gas. The oven temperature was programmed at 35 °C (for 5 min) to 280 °C at 10 °C min^−1^, and then held isothermally at 280 °C for 10.5 min. An aliquot of 1 μL of each oil (100 mg of each sample was dissolved in 10 mL of dichloromethane) was injected in the splitless mode (column effluent was split 1:1 for simultaneous detection). The ion source temperature was 230 °C; electron ionization mass spectra were acquired at 70 eV within a mass range of 38–550 Da (Da) during a scan time of 0.73 scans s^−1^. Compounds were identified using ChemStation software (Agilent) by comparison of mass spectral data of their retention time with library data: Adams and NIST 05. Identities of some constituents were confirmed by co-injection with commercially available authentic standards. Quantification was based on calibration curves (peak area vs. concentration) generated from authentic standards of identified compounds and by flame ionization gas chromatography (CG/FID), under the same conditions as in the GC/MS analysis.

### 2.4. O. kilimandscharicum Essential Oil Formulations

*O. kilimandschericum* essential oil formulations were prepared by mixing either with acetone or Tween 80 and water.

#### 2.4.1. Emulsified *O. kilimandscharicum* Oil Formulation

This was prepared through a low-energy titration method [39] by mixing 8.33%(*v*/*v*) of Tween 80 (polyethylene glycol sorbitan monooleate, a non-ionic surfactant, and oil-in-water emulsifier) (Sigma-Aldrich, St. Louis, MO, USA), 16.67% (*v*/*v*) of *O. kilimandscharicum* oil, and 75% (*v*/*v*) of water. To achieve a stable emulsion with a hydrophilic–lipophilic balance (HLB), values above 12 were targeted. This was achieved through the use of Tween 80 with an HLB of 15. The essential oil and surfactant Tween-80 were pooled together and homogenized by stirring for 30 min at 700 rpm in a magnetic stirrer (Fisatom, Brazil). Then, water was added at a controlled flow rate of approximately 4 mL/min and stirred for 1 h. The final emulsion concentrate containing 16.67% of *O. kilimandscharicum* essential oil was made up to 1 litre with the addition of distilled water to obtain a 1000 ppm stock solution that was diluted to various concentrations for use in the larvicidal bioassay. The 8.33% (*v*/*v*) Tween 80 dissolved in water was used as the control.

#### 2.4.2. *O. kilimanscharicum* Essential Oil in a Solvent

*O. kilimanscharicum* essential oil dissolved in an organic solvent (acetone) was prepared following the WHO method [40]. Stock solutions of the oils in acetone were prepared at a concentration of 50 mg/mL, and then further serially diluted with acetone to six different concentrations. An appropriate amount of each of the stock solutions contained in shallow beakers was further diluted with distilled water to make 100 mL of the final test solutions with concentrations ranging from 0.1–1.0 ppm and an acetone content not exceeding 1% *v*/*v*. A 1% *v*/*v* aqueous acetone solution served as a control.

### 2.5. Sources of Mosquito Larvae

The experiments were carried out with larvae of *An. gambiae* s.s and *An. arabiensis* mosquitoes all from colonies maintained at the International Centre of Insect Physiology and Ecology (ICIPE) Insect Mass Rearing and Containment Unit. Larvae for each species were reared separately under laboratory conditions where the water temperature was maintained at 28 ± 2 °C throughout larval development. Hatched larvae were transferred to larger pans (37 cm × 31 cm × 6 cm) at densities of 200–300 larvae per pan at the 2nd instar and fed on Tetramin^®^ fish food (Tetra GmbH, Melle, Germany). At the 3rd instar, they were ready to be used in the experiment. The rearing water was replaced with fresh water and diet after every two days. The pupae were held in plastic cups and transferred into standard 30 cm × 30 cm × 30 cm rearing cages where they hatched into adults.

### 2.6. Larvicidal Activity of O. kilimandscharicum Oil

*O. kilimandscharicum* oil dissolved in acetone and the emulsified *O. kilimandscharicum* oil formulation was evaluated for larvicidal activity against mosquito larvae. The tests were conducted in the laboratory following the standard WHO method, with some modification [40]. At first, mosquito larvae were subjected to a wide range of concentrations of the two formulated solutions and their respective controls to determine their activity profiles. Four aliquots from the stock solutions were prepared using the serial dilution method to give 4 concentrations (1, 0.5, 0.25, and 0.1 ppm), which gave 10% to 95% mortality in 24 h or 48 h. The concentrations were subsequently applied to establish the lethal concentration of fifty percent (LC_50_) and ninety percent lethal concentration (LC_90_) values of the oil and aqueous formulation. Groups of 25 (*n* = 25) late third instar larvae were transferred through droppers to 200 mL beakers, containing 100 mL of the test blends with various doses. All the experiments were performed in triplicate with an equal number of controls that were set up simultaneously. Two controls were set up: one with acetone–Tween 80 in dechlorinated tap water and one positive control with *Bti*. (Under field-simulated evaluation). Each test was run three times on different days. Larval food Tetramin^®^ in fish food (Tetra GmbH, Melle, Germany) was added to each test beaker using a dipstick (approximately 0.02 g) to enable long-term observations. The food was supplemented every 48 h and test blends were topped up with water up to 100 mL to ensure the test sample did not change. The test beakers were maintained at 25–28 °C and a photoperiod of 12 h light followed by 12 h darkness (1:12D). Larval mortality was scored after 24 h and 48 h of exposures. Larvae were considered dead if they remained irresponsive within two minutes when gently probed with a pipette. If more than 10% of the control larvae developed into pupae during the experiment, the experiment was abandoned and repeated because late instar larvae do not ingest 24 h before pupation, and too many larvae might have endured merely because they were too old. For cases where mortality was between 5 and 20%, the moralities of treated units were corrected using Abbott’s formula [41] as indicated in Equation (1).
(1)$$\mathrm{Mortality}(\%)=\frac{\left[X-Y\right]}{X}100$$ where *X* is the percentage survival in the untreated control and *Y* is the percentage survival in the treated group.

### 2.7. Effects of Sublethal Doses of Emulsified O. kilimandscharicam Oil Formulation

To determine the effects of sublethal exposure to emulsified *O. kilimandscharicum* oil formulation during larval development, beakers with third instar larvae were obtained from the insectary and were exposed to the above experimental conditions for 48 h. These experiments were replicated 4 times. For each experiment, four controls were used containing the sterilized water and four treatments for each of the three sublethal emulsified *O. kilimandscharicum* oil formulations concentrations (LC_20_, LC_50_, and LC_70_). During the 48-h exposure period, each beaker was provided with 20 mg of Tetramin^®^ in fish food (tetramine) per day. All dead and moribund larvae were counted after 24 and 48 h. After the 48-h exposure period for each replicate, all surviving larvae from the same treatment were pooled and placed in new breakers with fresh salt-treated water only. All emerging pupae were placed in plastic cups (100 × 50 mm diameter), which were placed in 30 × 30 × 30 cm cages, separated by treatment and replicate. The emerging adults were fed 6% sugar solution ad libitum. Under the microscope, morphological defects were monitored at intervals of 24 h until the death of the last larva or emergence of an adult.

### 2.8. Larvicidal Activity under Field-Simulated Conditions

Larvicidal activity of the emulsified *O. kilimandscharicum* oil formulation was evaluated under field-simulated conditions using natural mosquito breeding sites in Jaribuni village (03°37.3′ S; 039°44.6′ E), Kilifi county in Kenya, in September and in October during the short rain season. The WHO mosquito larvicidal protocol was applied under the field-simulated trial method [40]. Multiple artificial 5 L clay pots were used in the field. The experiment was carried out using 3rd instar *An. gambiae* larvae which were reared in the laboratory. The pots were filled with 1 kg of sterilized soil and 3 L of water from the mosquito breeding habitat. The set setup was left for ~24 h and then batches of 50 *An. gambiae* s.s. larvae were introduced in each pot. After 2 to 3 h of larval acclimatization, the containers were treated with 0.1, 0.5, and 1 ppm of emulsified *O. kilimandscharicum* oil formulation concentrations based on the volume of water in the pot using perforated jars. *Bti* granules, VectoBac WDG/WG-3000 ITU, were used as positive control with a concentration of 28 ppm, i.e., the concentration of 4 *Bti* granules per 10 cm × 10 cm surface area, as recommended in the WHO Larval Source Management protocol [42]. The *Bti* granules were broadcasted in a randomized manner over the water surfaces. The pots were covered with nylon mesh screens to prevent other mosquitoes or other insects from laying eggs, and to protect the water from falling debris. The water level in each pot was maintained by topping up as necessary. Five replicates of each dosage of the essential oil formulation were used. *Bti* granules were used as the positive control. The larvae in the pots were assessed after 24 h and 48 h, whereby live ones were counted and recorded. For low-dosage sample materials, the persistence of the larvae, pupae, and pupal skins was assessed for 7 days. Beyond this time, all larvae had pupated and developed into adults. pH and water temperature values were noted during the evaluation period. 

### 2.9. Data Analysis

The mortality rate was expressed as % mean ± S.D of experimental replicates for each dosage of the test solutions after correction using Abbotts’s formulae based on the data obtained from the negative control. Log probit analysis [41] was used to determine the LC_50_ and LC_90_ at their associated 95% fiducial limits at upper and lower confidence limit (UCL/LCL) using R software version 3.2.3 [43]. A comparison of LC_50_ and LC_90_ values was based on confidential limits for each population, whereby there was no significant difference if two confidence intervals overlapped. The data obtained were analyzed using descriptive statistical analyses. Means were compared by measured ANOVA followed by the post-hoc Tukey HSD test and *p* values of less than 0.05 were considered statistically significant. Graphs were designed using Graph Pad Prism version 7.01 for Windows (GraphPad) Software, San Diego, CA, USA).

## 3. Results

### 3.1. Oil Yield of O. kilimandscharicum

The average percentage yield of the essential oil from dried aerial parts of *O. kilimandscharicum* was 3.51 ± 0.56% during the wet season and 3.68 ± 0.33% during the dry season. The two seasons that prevail in Kenya are the dry season (December–March and July–October) and wet season (April–June and November), which both impact the oil yield. Through the seasons, the oil yield obtained in the wet season was greater than that harvested in the dry season, although not significantly different. The oil had a relative density of 0.98, which is lower than that of water. The oil color was light yellow.

### 3.2. Chemical Composition of the Essential Oil

Thirty-nine major compounds identified in *O. kilimandscharicum* oil. D-camphor was the most abundant constituent (36.6%), followed by limonene (18.6%), camphene (7.1%), linalool (4.3%), terpinen-4-ol (3.9%), *α*-terpineol (3.6%), terpinolene (2.2%), and sesquisabinene (2.1%) (Table 1).

Retention indices (RIs) were experimentally determined against C_5_–C_18_ n-alkanes using the non-polar HP-5MS capillary column (30 m × 0.25 mm i.d.; 0.25 μm film thickness, with 5% phenylmethyl silicone as the stationary phase; J & W Scientific, Folsom, CA, USA). The content was expressed as percentages obtained by integrating the gas chromatography peak area in the *O. kilimandscharicum* essential oil. RI: retention index. Conditions of analysis involved gas chromatography associated with mass spectrometer (GC-MS) THERMO DSQ II. Chemical constituents were identified by comparison of the mass spectra obtained with published spectra [44]. 

### 3.3. Laboratory Larvicidal Bioassay

The concentration of *O. kilimandscharicum* oil dissolved in acetone which induced the median larvicidal activity (LC_50_) after 24 h and 48 h against third instar larvae of *An. gambiae* was 0.74 and 0.31 ppm, respectively (Table 2). On the other hand, the median larvicidal potency (LC_50_) of the emulsified *O. kilimandscharicum* oil formulation after 24 h against third instar larvae of *An. gambiae* and *An. arabiensis* was 0.74 and 3.45 ppm, and the LC_90_ was 2.39 and 4.72 ppm, respectively (Table 2). After 48 h, the LC_50_ was 0.14 ppm for both against the third instar larvae of *An. gambiae* and *An.*
*arabiensis*, while the LC_90_ was 0.22 and 1.63 ppm, respectively (Table 2).

From the results, emulsified *O. kilimandscharicum* oil formulation significantly exhibited higher larvicidal activity in comparison to the oil dissolved in acetone. On the other hand, the difference in larvicidal activity of the emulsified *O. kilimandscharicum* oil formulation was significantly different from the nonionic surfactant and emulsifier Tween 80. The LC_50_ of emulsified *O. kilimandscharicum* oil formulation was 0.07 ppm while the aqueous non-ionic surfactant was at 172.57 ppm after 24 h exposure against *An. gambiae* larvae. This was a quite very low activity for Tween 80 which gave high LC_50_ values. As such, Tween 80 did not have any significant effect on larvae mortality (*p* > 0.05). Furthermore, 100% larval persistence was observed in the untreated (water only) control group for the whole experiment cycle.

### 3.4. Effects of Sublethal Doses of the Emulsified O. kilimandscharicum Oil Formulation

It was observed that sublethal doses of emulsified *O. kilimandscharicum* oil formulation caused low mortality (Table 3).

The number of mosquitoes larvae that developed to the adult stage decreased with increasing concentrations of emulsified *O. kilimandscharicum* oil formulation (Table 3). Most of the larvae in the treatment experiment revealed growth disruption larvae, as demonstrated in Figure 1. Figure 1a,e display a normal *An. gambiae* s.s and *An. arabiensis* larval–pupal intermediary, respectively, noted in the negative control. The abnormally increased development rate of *An. gambiae* larvae led to insufficient myelinization, as revealed in Figure 1b, whereas during the same time, stunted *An. arabiensis* larval–pupal intermediate was observed in Figure 1c,f,g which were subjected to a lower dose of the formulation, showing uncharacteristic abnormal and dead larval–pupal intermediate of *An. gambiae* and *An. arabiensis*, respectively. Despite molting being sustained, the growth of the infantile phases was extensively impacted. The microscopic examination of the dead immature phases at 30× enlargement showed morphologic deficiencies. This was manifested similar to abnormal dead larval pupal intermediates in *An. gambiae*, as well as crumbled mouthparts and wings within the pupal exuvial, as shown in Figure 1d. The adults that emerged from the lower dose application were incapable of breaking from the pupal caste and perished on the surface of the test solution for the case of *An. arabiensis*, as shown in Figure 1h. Generally, the formulation at lower doses caused a prolonged larval phase period of four days before pupation, unlike the negative control.

### 3.5. Field-Simulated Evaluation

The results of the field-simulated evaluation are illustrated in Figure 2. The emulsified *O. kilimandscharicum* oil formulation had great efficacy compared with *Bti* granules used as a positive control. After 1 day of treatment, the formulation at 0.5 ppm led to 98% larval mortality, whereas *Bti* at 28 ppm exhibited larval mortality of 54%. When the emulsified *O. kilimandscharicum* oil formulation attained 100% mortality on day 3, *Bti* was at 76.5%, as indicated in Figure 2. The survival of larvae in the control remained high since there was low mortality and only started to decline when pupation commenced.

## 4. Discussion

Even though there has been a recent decline in the global prevalence of some vector diseases, such as malaria, the vector diseases remain to be responsible for high morbidity and mortality in sub-Sahara Africa [45]. There is the necessity to seek alternative ways to combat the menace particularly in the Africa continent [46]. In this study, we explored the use of *O. kilimandscharicum* essential oil as a source of an effective mosquito larvicide.

There was a difference in the oil yield of *O. kilimandscharicum* which could be attributed to the two weather seasons when the samples were collected. The two-weather condition that prevails in Kenya concerns the dry season (December–March and July–October) and the wet season (April–June and November). Through the seasons, the oil yield obtained in the wet season was greater than that harvested in the dry season, although not significantly different. The major chemical composition of *O. kilimandscharicum* concerns camphor, 1,8-cineole, linalool, and limonene, as reported in previous studies [33,47].

The LC_50_ for the unformulated oil and that of the emulsified *O. kilimandscharicum* oil was LC_50_ = 0.74 and LC_50_ = 0.07 ppm, respectively. Our results suggest that the *O. kilimandscharicum* essential oil bioactivity was enhanced to a more effective larvicide when formulated. The LC_50_ for the emulsified *O. kilimandscharicum* oil formulation was 0.14 ppm against *An. arabiensis.* The comparable improved performance was obtained with neem oil after formulation [46], whereby the improved lethal concentration (LC_50_) of the neem water emulsion formulation against *An. stephensi*, *Cx. quinquefasciatus* and *Ae. aegypti* was reported to as 1.6, 1.8, and 1.7 ppm, respectively [48]. Moreover, our findings indicate that higher doses were toxic to the larvae while sublethal doses of the formulation reduced their life spans. Similar results have been reported for *Culex quinquefasciatus* when exposed to sublethal doses of cypermethrin for both larvae and adults [49]. The toxic effect was attributed to physiological damage caused to the nervous system and associated aberrations due to abnormal hormone release and dehydration because of exposure to cypermethrin. In this study, it was observed that, at sublethal doses, the emulsified *O. kilimandscharicum* oil formulation induced developmental disruptions, as indicated in Figure 1. The bioactivity was attributed to interruption of the endocrine system stability by essential oils and their constituents which disturb their biochemical processes. The instability may result from neurotoxicity and capacity to act as insect growth regulators, disrupting the ordinary process of morphogenesis [50,51]. The bioactivity may be associated with the presence of monoterpenoids and sesquiterpenes, including camphor, D-limonene, myrcene, terpineol, linalool, and pulegone, as reported in Table 1, which acted in synergy. The compounds have also been previously assessed for their neurotoxicity against house flies and German cockroaches [52]. Camphor, the major constituent of the essential oil of *O. kilimandscharicum*, has similarly shown strong insecticidal bioactivity against postharvest insect pests [53]. Neurotoxicity by essential oils in insects is characterized by hyperactivity leading to rapid ‘’knock-down’’ and immobilization of the organism [54]. Furthermore, several studies have established that essential oils from aromatic plants inhibit acetylcholinesterase (AChE) in different insect species [55,56,57]. AChE plays a role in cholinergic synapses that are essential for insects and higher animals [58]. Inhibition of AChE leads to a high deposit of acetylcholine at the synapses. This puts the post-synaptic membrane in a state of perpetual stimulation, leading to ataxia, the wide-ranging deficiency of conformity in the neuromuscular system, and even death [59,60]. On the other hand, the observed stunted growth and growth disruption among the larvae under a low dose of the formulation of *O. kilimandscharicum* oil could be attributed to averting feeding through chemoreception, possibly due to volatile organic constituents and the disruption of endocrine functions, causing regulated insect growth and, thus, triggering developmental irregularities [61]. The larvae stage is critical to the life of an insect and is susceptible to specific toxic effects on specific enzyme systems as well as disturbances in hormonal triggering [62]. Furthermore, larvae exposure to sublethal doses of emulsified *O.*
*kilimandscharicum* oil formulation could lead to a reduction in vectorial capacity for disease vector populations, following the reduced lifespan of subsequent adults [63].

Under the simulated-field trial, the emulsified *O. kilimandscharicum* oil formulation exhibited significant larvicidal activity within the first 24 h as well as *Bti* granules used as a positive control, as illustrated in Figure 2. Given that temephos and *Bti* are currently the most used larvicidal agents to fight mosquito larvae, with the challenge of larvae resistance to temephos and the high cost of *Bti* production on a large scale, natural products such as *O. kilimandscharicum* oil present alternative larvicide sources when properly formulated. Moreover, several phytochemicals, particularly from the Lamiaceae family, have been found to possess larvicidal activity against larvae of *Ae. aegypti*, *An. gambiae*, *An. arabiensis*, and *Cx. quinquefasciatus*, including *Lippia alba*, *O. sanctum*, *O. gratissimum*, and *O. basilicum* [38,39,60,64]. In addition, recently EO from *Origanum vulgar* has shown good larvicidal activity against *Ae. aegypti* at 20 ppm before formulation into water emulsion [63,64]. The use of this plant’s EO as larvicide has been limited due to the complication of immiscibility in water and the loss of activity due to high volatility [65]. However, phytochemical-combined formulations can improve activity and improve delivery for effective vector control [66]. As such, the present study lays the useful groundwork for the practical application of essential oil from *O. kilimandscharicum* as a mosquito larvicidal product. The bioactive formulation has the advantage of high solubility in aqueous media; it is also easily produced, ecofriendly, and affordable. However, given that the plant grows widely in different agro-ecological areas in tropical Africa, and for effective downstream applications, it will be helpful to compare the composition (epigenetic/chemotypes variations) and larvicidal effects of essential oil of the plants growing in different locations. In addition, it will also be helpful to evaluate essential oils from other plants with similar bioactivity for blending or ensuring sustainability. The plant world provides a rich untapped pool of phytochemicals that may offer alternatives to synthetic insecticides in mosquito control programs [64,67]. Moreover, for long-term efficacy and deployment, there is a need to evaluate the contribution of each of their components on the bioactivity of each blend and assess their resistance-mitigating effects [68]. In addition, the effects on the nontarget organism in the aquatic environment need to be determined in detail.

## 5. Conclusions

The emulsified *O. kilimandscharicum* oil formulation exhibited significant larvicidal activity against *An. gambiae* and *An. arabiensis* larvae. Despite its direct toxicity to the larvae, its significant sublethal effects were an additional hallmark to demonstrate the further activity of this plant extract through morphological and physiological anomalies, making emulsified *O. kilimandscharicum* oil formulation a potential candidate to be used as a new plant-based insecticide to control disease vectors. Moreover, due to the fact that *O. kilimandscharicum* plant has high EO yields and can be easily and widely cultivated, it is a valuable alternative for the effective and eco-friendly control of disease vectors for both developing and developed communities.

## Figures and Tables

**Figure 1 insects-13-00203-f001:**
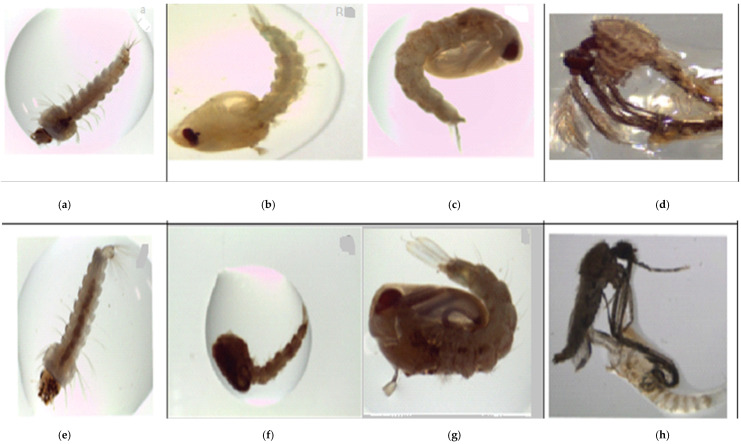
Effect of sub-lethal dose of emulsified *O. kilimandscharicum* oil formulation on mosquito larval development, pupation, and adult stage. (**a**) Normal *An. gambiae s.s*. (**b**) Demelanized *An. gambiae* s.s larval-pupal intermediate. (**c**) Abnormal *An. gambiae* s.s larval–pupal intermediate. (**d**) Arrested adult emergence in *An.gambiae*. (**e**) Abnormal *An. arabiensis* larvae in negative control solution. (**f**) A stunted *An. arabiensis* larval–pupal intermediate. (**g**) Abnormal *An. arabiensis* larval–pupal intermediate. (**h**) Failed adult emergence in *An. Arabiensis*.

**Figure 2 insects-13-00203-f002:**
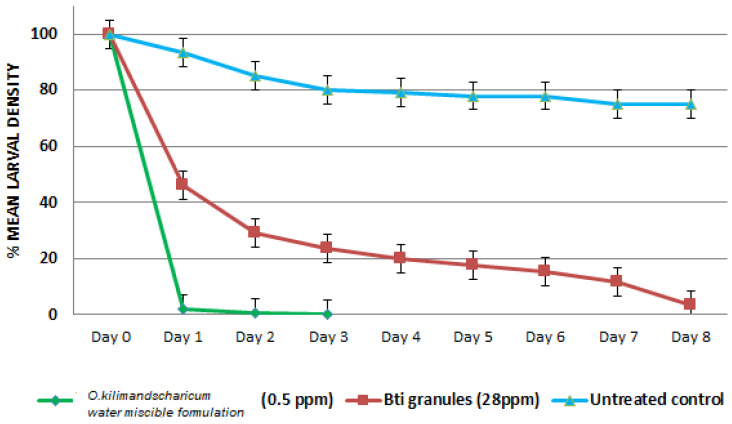
Larvicidal activity under field-simulated conditions in Kilifi of emulsified *O. kilimandscharicum* oil formulation and *Bti* granules.

**Table 1 insects-13-00203-t001:** Chemical composition of the essential oil of *O. kilimandscharicum* harvested in the dry season.

RI	Compounds	Relative Content (%)
923	Tricyclene	0.71
931	*α*-Pinene	1.94
946	Camphene	7.14
989	Myrcene	1.88
1003	*α*-Phellandrene	0.76
1008	*δ*-2-Carene	0.68
1024	*p*-Cymene	0.23
1029	Limonene	18.61
1046	(*E*)-*β*-Ocimene	0.92
1059	*γ*-Terpinene	1.05
1085	*α*-Terpinolene	2.23
1093	6-Camphenol	0.6
1098	*Trans*-sabinene hydrate	0.86
1100	Linalool	4.32
1143	Camphor	36.58
1158	Isoborneol	0.14
1165	Borneol	0.97
1167	*α*-Terpineol	3.6
1180	Terpinen-4-ol	3.9
1247	Geranial	0.21
1286	(*E*)-Linalool oxide acetate	1.4
1351	*α*-Cubebene	0.08
1357	Eugenol	0.14
1378	*α*-Copaene	0.5
1419	(*E*)-*β*-Caryophyllene	1.94
1433	*β*-Copaene	1.07
1450	*trans*-Muurola-3,5-diene	0.26
1455	*α*-Humulene	0.38
1470	Bicyclogermacrene	0.47
1476	Geranyl propanoate	1.28
1480	*γ*-Muurolene	0.12
1482	Germacrene D	1.17
1497	Viridiflorene	0.46
1503	*α*-Bisabolene	0.12
1541	Sesquisabinene hydrate, cis-	2.12
1601	Humulene epoxide II	0.08
1700	Geranyl propanoate	0.17
1470	Dauca-5,8-diene	0.23

**Table 2 insects-13-00203-t002:** Mortality level (%) of 3rd instar mosquito larvae.

Treatment	Mosquito Species	Time (h)	Concentrations (ppm)/% Mortality					Lethal Concentration		Treatment Response	
			Control (-ve)-Water	0.1	0.25	0.5	1	LC_50_ (95%CL)	LC_90_ (95%CL)	R^2^	X^2^, df, *p*-Value
*O. kilimandscharicum* oil in acetone	*An. gambiae*	24	0.0	4.7	18.2	37.1	60.0	0.74 (0.07–0.97)	3.45 (2.13–8.73)	0.9929	X^2^ = 1954.00 df = 18 *p* < 0.0001
		48	0.0	20.4	43.6	63.4	100.0	0.31 (0.11–0.87)	1.82 (0.06–51.63)	0.9786	X^2^ = 2025.06 df = 4, *p* < 0.0001
*O. kilimandscharicum water-emulsion*	*An. gambiae*	24	0.0	34.2	67.2	75.9	99.4	0.07 (0.61–0.13)	2.39 (1.08–19.70)	0.794	X^2^ = 1923.69 d 8, *p*< 0.0001
		48	0.0	21.0	94.0	99.9	100.0	0.14 (0.09–0.20)	0.22 (0.10–0.49)	0.9394	X^2^ = 794.00, Df = 4, *p* < 0.0001
	*An. arabiensis*	24	0.0	29.9	46.2	59.0	71.0	0.31 (0.18–0.50)	4.72 (1.74–117.50)	0.87635	X^2^ = 4688.80 df = 18, *p* < 0.0001
		48	0.0	43.2	62.1	74.8	84.8	0.14 (0.07–0.20)	1.63 (0.86–7.52)	0.8394	X^2^ = 3052.08 d 8, *p* < 0.0001
Emulsion/surfactant (Tween 80)	*An. arabiensis*	24	0.0	0.4	1.1	2.3	4.3	75.28 (10.70)	1877.55 (69.99)	0.9978	X^2^ = 47.97 df = 18, *p* < 0.0001
		48	0.0	0.9	2.3	4.3	7.7	30.162 (8.63–1145.80)	648.88 (68.05–475,024.38)	0.9973	X^2^ = 75.11 df = 18, *p* < 0.0001
	*An. gambiae*	24	0.0	1.3	2.6	4.1	6.3	172.57 (14.78–0.00)	12,795.98 (172.20	0.9694	X^2^ = 111.91 df = 18, *p* < 0.0001
		48	0.0	1.6	3.6	6.1	9.9	31.99 (8.13–615.21)	1011.90 (78.57–3,997,163.24)	0.9847	X^2^ = 142.31 df = 18, *p* < 0.0001

LC_50_ = Mean lethal concentration, 95% CL = upper/lower confidence Interval. Comparison of LC_50_ and LC_90_ values based on confidential limits in case C.L. Overlap showed no significant differences at *p* < 0.05. The emulsified *O. kilimandscharicum* oil formulation showed comparable levels of activity against third instar larvae of *An. gambiae* and *An. arabiensis*, as shown in Table 3. Compared to negative controls, the LC_50_ for emulsified *O. kilimandscharicum* oil formulation was the lowest and substantially reduced survival rates of *An. gambiae* and *An. arabiensis.* The LC_50_ for the formulation was highest against *An. gambiae* larvae. There was considerable vulnerability variation amongst the two mosquito species to the formulation based on the degree of overlap of fiducial limits (FL/CL, *α* = 0.05, *p* = 0.05).

**Table 3 insects-13-00203-t003:** Effect of larval exposure to sublethal doses of emulsified *O. kilimandscharicum* oil formulation.

Treatment (ppm)	*An. gambiae*		*An. arabiensis*	
	Mean No of Pupae (M ± SE)	Mean No of Adults (M ± SE)	Mean No of Pupae (M ± SE)	Mean No of Adults (M ± SE)
LC_70_	31.3 ± 1.68 a	29.0 ± 2.3 a	32.0 ± 1.15 a	22.3 ± 1.28 a
LC_50_	49.7 ±1.68 b	44.7 ± 2.3 b	45.0 ± 1.15 b	39.0 ± 1.28 b
LC_20_	71.3 ± 1.68 c	67.0 ± 2.3 c	65.0 ± 1.15 c	62.3 ± 1.28 c
Control	79.0 ± 1.68 d	75.3 ± 2.3 d	73.0 ± 1.15 d	68.0 ± 1.28 c

Number (mean ± SE) of pupae and adults emerged from 100 larvae exposed to sublethal doses of emulsified *O. kilimandscharicum* oil formulation. Means followed by the same letter are not significantly different in a Tukey test, at *p* < 0.05.

## Data Availability

All data generated or analyzed during this study are included in this manuscript.

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
