# Peer review of "Mosquitoes Larvicidal Activity of Ocimum kilimandscharicum Oil Formulation under Laboratory and Field-Simulated Conditions"

_insects, 2022, doi:10.3390/insects13020203_

Round 1
Reviewer 1 Report
Dear authors,
This paper deals with the evaluation of larvicidal effects of plant derived essential oil from Ocimum kilimandscharicum and its water-based formulation against 4 mosquito species.
Further down, you will find some comments of major importance that need appropriate reply by the authors to consider the suitability of the manuscript for publication.
Line 23: Non target organisms were not tested, so this statement should be removed.
Lines 110-113: You could elaborate more on the scope of the manuscript indicating the key steps of the current research, i.e. i) evaluation of larvicidal potential of Ocimum kilimandscharicum essential oil against An gambiae ii) efficacy evaluation of larvicidal effect of a water based formulation of this oil against 4 mosquito species, and iii) the performance of the water-based formulation was tested against An. gambiae in semi-field conditions.
Lines 148-149: The use of a vegetable oil as negative control seems not clear to me and not relevant. What is the composition of this oil? What is the relationship with the tested oil? Normally, as untreated negative controls, acetone solution and emulsifier solution in water at the tested concentrations should have been used. Please clarify. It is necessary to supplement the current work with additional bioassays with appropriate untreated negative controls against all tested mosquito species.
Line 167: It should be clarified that the acetone solution was tested only against An. gambiae, whereas the aqueous formulation against all tested species.
Lines 181-183: When food was added? After 48h? please clarify in the text.
Lines 185-187: Did you transfer the treated larvae to clean water with food? Did you add water into the beakers to ensure that the concentration did not change? Please clarify against which species the sublethal doses were applied.
Line 202: How much water did you place in each pot?
Line 208-209: How did you choose this particular dose of B.t.i? Is it the authorized one? Please justify in the text.
Table 3 (and results in 3.3): Comparison of LC50 and LC90 values should be based on confidential limits, i.e. in case C.L. fail to overlap no significant differences are observed. This should be added as footnote. The x2 and d.f. values from probit analysis should also be added in the table. The cases where a heterogeneity factor is calculated for the C.L. this should also be indicated in the table. I propose not to compare means with ANOVA and post hoc test. In the text you should not refer to median concentrations, but just to LC50 and LC90 values.
In the acetone water-based solution of the essential oil against An. gambiae, the lower limit is zero (0) for the LC50 and LC90 values. How is it possible? Please check. Does it mean that the acetone solution (control) was toxic? Please double check the results of the analysis and revise the table and the text accordingly.
Lines 284-302: The results of the sublethal effects, i.e. the % of survival to pupa and adulthood and larval and pupal stage longevity, could be presented in a table and appropriately discussed.
Table 4: Mortalities among the essential oil, Bti and the control should be statistically compared by a parametric test (e.g. ANOVA followed by a post hoc test) or a non-parametric test. In the table and in the text, it should be clarified that the results are referring to larvicidal activity against An. gambiae.
Line 313: It should be 77%, not 96,5% as stated.
Line 348: “… a more effective larvicide against An. gambiae as….”
You should elaborate more on the discussion of larvicidal effects of the tested essential oil from Ocimum against the 4 mosquito species using relevant literature data with bioassays with essential oils or formulations based on other Lamiaceae or Ocimum plants against the tested mosquito species.
Author Response
Point 1: Line 23: Non target organisms were not tested, so this statement should be removed.
Response 1: Statement removed
Point 2: Lines 110-113: You could elaborate more on the scope of the manuscript indicating the key steps of the current research, i.e. i) evaluation of larvicidal potential of Ocimum kilimandscharicum essential oil against An gambiae ii) efficacy evaluation of larvicidal effect of a water based formulation of this oil against 4 mosquito species, and iii) the performance of the water-based formulation was tested against An. gambiae in semi-field conditions.
Response 2: Scope of study elaborated as below
This study outlines results of the evaluation of the larvicidal potential of O. kilimandscharicum essential oil against An. gambiae and present the use of a low energy nano-emulsion delivery system with its volatile oil and evaluated its larvicidal activity against 4 mosquito species under laboratory conditions. In addition, the performance of the water-based formulation against An. gambiae in semi-field conditions was assessed
Point 3: Lines 148-149: The use of a vegetable oil as negative control seems not clear to me and not relevant. What is the composition of this oil? What is the relationship with the tested oil? Normally, as untreated negative controls, acetone solution and emulsifier solution in water at the tested concentrations should have been used. Please clarify. It is necessary to supplement the current work with additional bioassays with appropriate untreated negative controls against all tested mosquito species.
Response 3: We have opted to drop vegetable oil and adopted the aqueous polysorbate 80 (a nonionic surfactant and emulsifier (Tween80) as the control. A bioassay experiment was set up to test the Tween80 against two of the mosquito species (An. gambiae and An. arabiesis) See table 3. The other two were not tested due to unavailability within the short time.
Point 4: Line 167: It should be clarified that the acetone solution was tested only against An. gambiae, whereas the aqueous formulation against all tested species.
Response 4: This has been clarified and emphasised in the text and tables 3
Point 5: Lines 181-183: When food was added? After 48h? please clarify in the text.
Response 4: Food was added immediately after the transfer of the larvae but was replenished only after 48h for cases of longer observation
Point 5: Lines 185-187: Did you transfer the treated larvae to clean water with food? Did you add water into the beakers to ensure that the concentration did not change? Please clarify against which species the sublethal doses were applied.
Response 5: The treated larvae remained in the same beaker, whereas water was topped up to original level in the beaker to ensure there was no change of concentration
Point 6: Line 202: How much water did you place in each pot?
Response 6: The 5 liter pots were filled with 3 litres and ensured a depth of 5-10 cm
Point:7 Line 208-209: How did you choose this dose of B.t.i? Is it the authorized one? Please justify in the text.
Response 7: The 28ppm correspond to achieving approximately 4 granules per 10 cm x10 cm surface area as recommended in the WHO Larval Source Management protocol (WHO, LSM, 2013)
Point :8 Table 3 (and results in 3.3): Comparison of LC50 and LC90 valuess hould be based on confidential limits, i.e. in case C.L. fail to overlap no significant differences are observed. This should be added as footnote. The x2 and d.f. values from probit analysis should also be added in the table. The cases where a heterogeneity factor is calculated for the C.L. this should also be indicated in the table. I propose not to compare means with ANOVA and post hoc test. In the text you should not refer to median concentrations, but just to LC50 and LC90 values.
Response 8: Comparison of LC50 and LC90 values have been based on confidential limits and significant difference or lack of it shown by letters.
The foot note”’ Comparison of LC50 and LC90 values should be based on confidential limits in case C.L. fail to overlap no significant differences observed at p ≤ 0.05. (Student-Newman-Keuls test),’’ has been added at the bottom of the table
Point 9: In the acetone water-based solution of the essential oil against An. gambiae, the lower limit is zero (0) for the LC50 and LC90 values. How is it possible? Please check. Does it mean that the acetone solution (control) was toxic? Please double check the results of the analysis and revise the table and the text accordingly.
Response 9: These were figures generated in preliminary studies during range finding. This has been corrected in the Table 3. No lower confident limits read zero.
Point 10: Lines 284-302: The results of the sublethal effects, i.e. the % of survival to pupa and adulthood and larval and pupal stage longevity, could be presented in a table and appropriately discussed.
Response 10: This has been implemented as suggested. A table (4) listing the %larval survival at sub-lethal dose was generated. This is complimented by plate 1 which gives pictorial outlook of the various stages of larval growth.
Point 10: Table 4 (Now Table 5): Mortalities among the essential oil, Bti and the control should be statistically compared by a parametric test (e.g. ANOVA followed by a post hoc test) or a non-parametric test. In the table and in the text, it should be clarified that the results are referring to larvicidal activity against An. gambiae.
Response 9: The data was found to be not normally distributed hence analysis using ANOVA was not possible. The percentage means and their corresponding standard errors were generated using a Glm model (survival) and a non-parametric analysis (Tukey) was used to compare the means.
Point 11: Line 313: It should be 77%, not 96,5% as stated.
Response 11: Well, noted and corrected
Point 12: Line 348: “… a more effective larvicide against An. gambiae as….”
Response 12: Noted and corrected in the text
Point 13: You should elaborate more on the discussion of larvicidal effects of the tested essential oil from Ocimum against the 4 mosquito species using relevant literature data with bioassays with essential oils or formulations based on other Lamiaceae or Ocimum plants against the tested mosquito species.
Response 13: Noted and incorporated in the text
Several phytochemicals particularly from the lamiacea family have been found to possess larvicidal activity against larvae of Ae. aegyptei, An. gambiae, An. Arabiensis and C. quinquefasciatus including Lippia alba, O. sanctum, O. gratissimum and O. basilicum(39), (59). In addition, recently EO from Origanum vulgar have been shown to give good larvicidal activity against Ae. aegyptei at 20 ppm before conversion into water emulsion (61). Unfortunately, the use of this plants EO as larvicide has been limited due to the complication of immiscibility in water and loss of activity due to high volatility (Ferreira, 2019). However, phytochemical-combined formulations can, improve activity and improve delivery for effective vector control.
Submission Date
09 November 2021
Date of this review
15 Nov 2021 11:18:22

Reviewer 2 Report
The present manuscript describes the application of essential oils for the management of mosquito larvae. The findings are interesting and valuable to be published on Insects. There are some typos and English check.
Abstract and summary are basically identical! Authors need to reformulate one.
Introduction is well developed and present well the problem.
Material and methods require clarifications! Authors need to provide more information on the formulation development and preparation, which is currently lacking.
Data presentation format should be improved in the Results section.
In the Discussion section I would like to see more comments on formulation development, type of chemicals and substrates are used to improve an essential oil-based formulation, in relation to the formulation presented in this study.
Below some comments (and specific comments are incorporated into the attached pdf):
Materials and Methods require clarification:
- what type of detector was coupled with the GC? MS or FID? because they are very different and they provide different type of informations.
- The formulation of the EO is not at all described. Authors need to provide a detailed description of the protocol for synthesizing the formula. Also, no mention of the selection of the presented concentration: 18.67%! How the authors selected this specific concentration?!
- What is the control for the formula? what about the control for the formula? The authors need to test everything but the active ingredient (the EO) as control for the formula.
- Authors need to present more clearly the testing approch and clarify that the oil alone and when combined in the formula were tested for larvicidal activity. No mention of control!
- I suggest to have a table with all the treatments and concentrations tested.

Author Response
Point 1: Abstract and summary are basically identical! Authors need to reformulate one.
Response 1: This has been done. See below
Simple Summary: Mosquitoes are vectors of many severe diseases notably malaria, yellow as well as dengue fever; and lymphatic filariasis. Vector control with synthetic chemical insecticides has been associated with resistance development and undesirable human and ecological effects. Ocimum kilimandscharicum formulation was evaluated for its larvicidal activity against third instar larvae of Anopheles gambiae s.s., Aedes aegyptei, Anopheles arabiensis and Culex quinquefasciatus in the laboratory. The O. kilimandscharicum water-miscible formulation was then compared with Bacillus thuringiensis (Bti) granules on An. gambiae larvae under simulated field trials. In the present study, a O. kilimandscharicum essential oil water miscible formulation exhibited broad spectrum significant larvicidal activity against An. gambiae, An arabiensis, Ae. aegyptei and C. quinquefasciatus with LC50 range of 0.31-0.85ppm at 24 h and 0.07-0.74ppm at 48h. The high bioactive suggest its high potential as a botanical larvicide for control of disease vectors. The bioactive formulation had the advantage of high solubility in aqueous media, easily produced, ecofriendly and a low cost bioproduct. Moreover, the fact that O. kilimandscharicum can easily widely be cultivated, has high EO yields provide a practical application in effective and eco-friendly control of disease vectors among the resource poor and developed communities.
Abstract: Mosquitoes are vectors of many severe diseases including malaria, yellow as well as dengue fever; and lymphatic filariasis. Use of synthetic chemical insecticides for mosquito control has been associated with resistance development and detrimental human and ecological effects. For safer alternative Ocimum kilimandscharicum water-miscible formulation was evaluated for its larvicidal activity. Oil composition was analysed on GC-MS. The formulations were evaluated against 3rd instar larvae of Anopheles gambiae s.s., Aedes aegyptei, Anopheles arabiensis and Culex quinquefasciatus, in the laboratory and later compared with Bacillus thuringiensis against An. gambiae under semi-field conditions. The formulation showed significant larval mortalities agaist the four mosquito species with LC50 of 0.31-0.85ppm at 24 h and 0.07-0.74ppm at 48h. Under semi-field trial, within 24 h, the formulation showed 98% mortality compared with 54% of B.ti. By day 3, it was at 100%, Bti at 96.5%. Sub-lethal doses of the formulation led to growth disruption of the larvae as a mechanism of bioactivity. 39 constituents were identified in the oil, the main ones being D-camphor (36.6%) and limonene (18.6%). The results show great efficacy of O. kilimandscharicum water miscible formulation on mosquito larvae and provide a practical application in effective and eco-friendly control of disease vectors.
Point 2: Introduction is well developed and present well the problem.
Response 2: Thanks
Point 3: Material and methods require clarifications! Authors need to provide more information on the formulation development and preparation, which is currently lacking.
Response 3: This has been addressed by providing the steps taken during formulation as detailed below
Nano-emulsion formulation was achieved through low energy titration method (40). The aim was to achieve optimal emulsion formulation containing O. kilimandschericum essential oil as dispersed phase. To achieve a stable emulsion a a hydrophilic-lipophilic balance (HLB) value above 12 was targeted. The aqueous polysorbate 80 (a nonionic surfactant and emulsifierTween80, with HLB of 15) was used in the formulation as emulsifier. The formulation was constituted by 50% (w/w) of water, 8.33% (w/w) of the surfactant Tween 80 and 16.6% (w/w) of O. kilimandschericum oil for final 50g concentrate. The essential oil and surfactant Tween 80 were pooled together and homogenized by stirring for 30 min at 700rpm. Then water was added at a controlled flow rate of approximately 4ml/ min and stirred for 1 hr. The final emulsion concentrate containing 16.67 % of O. kilimandschericum essential oil was diluted to various concentrations for use in the larvicidal bioassay.
Point 3: Data presentation format should be improved in the Results section.
Response 3: This has been enhanced by consolidating the tables and making clarification in the title and foot notes
Point 4: In the Discussion section I would like to see more comments on formulation development, type of chemicals and substrates are used to improve an essential oil-based formulation, in relation to the formulation presented in this study.
Response 4: The formulation is briefly discussed as below
In the study, a water-miscible formulation was developed with the aim of having an optimal emulsion formulation containing O. kilimandschericum essential oil as dispersed phase. In emulsions, the stabilization of the oil droplets in the aqueous phase required the use of surfactants whose hydrophilic-lipophilic balance (HLB) value property was considered, Polysorbate 80 (a nonionic surfactant and emulsifierTween80, is a surfactant, considered as food grade, with very stable HLBs of 15. As such, for this purpose, it was used as emulsifier with essential oil of O. kilimandscharicum leading to a stable emulsion that was used in the evaluation for larvicidal bioactivity.
Point 5: Below some comments (and specific comments are incorporated into the attached pdf):
Response 5: Various comments, typos and suggestion appropriately responded to in the text and highlighted by track changes
Point 6: Materials and Methods require clarification:
Point 6a) what type of detector was coupled with the GC? MS or FID? because they are very different,] and they provide different type of information.
Response 6a): For identification of compounds the GC coupled to MS, however during quantification based on calibration curves, a flame ionization chromatography GC/FID was applied. Hence the 2 detectors were applied but at various levels
Point 6b): The formulation of the EO is not at all described. Authors need to provide a detailed description of the protocol for synthesizing the formula. Also, no mention of the selection of the presented concentration: 18.67%! How the authors selected this specific concentration?!
Response 6b): The formulation is now described and discussed
Nano-emuslsion formulation was achieved through low energy titration method (40). The aim was toachieve optimal emulsion formulation containing O. kilimandschericum essential oil as dispersed phase. To achieve a stable emulsion a a hydrophilic-lipophilic balance (HLB) value above 12 was targeted. The aqueous polysorbate 80 (a nonionic surfactant and emulsifierTween80, with HLB of 15) was used in the formulation as emulsifier. The formulation was constituted by 50% (w/w) of water, 8.33% (w/w) of the surfactant Tween 80 and 16.6% (w/w) of O. kilimandschericum oil for final 50g concentrate. The essential oil and surfactant Tween 80 were pooled together and homogenized by stirring for 30 min at 700rpm. Then water was added at a controlled flow rate of approximately 4ml/ min and stirred for 1 hr. The final emulsion concentrate containing 16.67 % of O. kilimandschericum essential oil was diluted to various concentrations for use in the larvicidal bioassay.
The concentration 18.6% which instead is supposed to be 16.7%, the inaccuracy being a typo error. The 16.7% point at the active % ingredient (O. kilimandschericum essential oil) composition in the initial stable emulsion concentrate generated as describe in the above formulation.
Point 6c): What is the control for the formula? what about the control for the formula? The authors need to test everything but the active ingredient (the EO) as control for the formula.
Response 6c): The control for the formula is a nonionic surfactant and emulsifierTween80 and the untreated control. The 2 have been incorporated in the evaluation bioassay. See Table 3.
Point 6d): Authors need to present more clearly the testing approch and clarify that the oil alone and when combined in the formula were tested for larvicidal activity. No mention of control!
Response 6d). Testing was done for the oil and when the oil was combined in the formulation as an emulsion. The results of the testing are listed in table 3.
Point 6e):I suggest to have a table with all the treatments and concentrations tested
Response 6e). This has been effected in Table 3
Submission Date
09 November 2021
Date of this review
12 Nov 2021 03:06:56

Round 2
Reviewer 1 Report
Dear authors,
Thank you for your responses. Many of my comments have been appropriately addressed in the revised manuscript, however in the attached file you you will find some major open points to be further revised/clarified.

Author Response
Response to Reviewer 1 Comments Rd 2
Point 1: I propose to remove the data from the bioassays against Culex and Aedes since no untreated controls were used. The manuscript should be focused on the two tested Anopheles species.so please amend accordingly.
Response 1: Data related to Culex and Aedes have been removed from table 3 following the reason raised,
Point 2: The fact that ‘’the treated larvae remained in the same beaker ‘should be addressed in the text
Response 2: The comment is now well addressed in the text and Table 4. A detailed experimental method followed to test sublethal doses of LC20, LC50 and LC70 is outlined. The larvae after 48 h were transferred to clean beakers with esterized water. Further monitoring was done to record mortality and any biological or morphological changes on the larvae, puape and emerging adult mosquitoes.
Point 3: You don’t have to put letters. The footnote phrase ‘’ in case CL fail to overlap no significant differences are observed at p<0.005 is sufficient.
Response 3: This was well noted and has be addressed in the text.
Point 4: ANOVA and post hoc (S-N-K) test are not appropriate for the comparison of LD50 or LD90 values estimated by probit analyses. Please omit any reference for the post hoc S-N-K test and clarify if the comparison was based only on confidential limits calculated with probit.
Response 4 The data was calculated with probit hence reference to post hoc S-N-K has been omitted hence comparison was based only on confidential limits.

Reviewer 2 Report
I am pleased to see the improvement made by the authors.
I have found few typos and some wording that need to be adjusted (pdf attached with comments). Also make sure to do English check.

Author Response
Response to Reviewer 2 Comments Round2
Reviewer Point 1: I have found few typos and some wording that need to be adjusted (pdf attached with comments).
Response 1: The highlighted typos and wording have been adjusted
Reviewer Point 2: Also make sure to do English check
Response 2: English check done and appropriate adjustments implemented.

Round 3
Reviewer 1 Report
Dear authors,
Thank you for considering my comments. However, you should revisit statistical issues regarding tables 3, 4 & 5 as mentioned in the attached document.

Author Response
Please consider the following comments regarding point 3: Point 3: Significant differences are observed at p<0.005 is sufficient.
Response 3: This was well noted and has be addressed in the text.
considered statistically significant” makes no sense
As already mentioned, from probit analysis the cases where a heterogeneity factor is calculated for the C.L. should be indicated in the table e.g. by an asterisk. A post hoc test (SNK) is not used for this purpose. Please see carefully the report from probit analysis to address this result in the table. This comment is referring to table 3 only. Noted and effceted
Response Table 3: The letters have been removed and heterogeneity addressed.
Table 5: Why did you remove letters from means in table 5? Please clarify what statistics you performed in for means comparison in table 5, i.e. parametric (ANOVA followed by a post hoc test, e.g. SNK) or non parametric (e.g. Kruskal Wallis). Any significant *Mean values are not since significant differences are determined when P<0.05.
Response Table 5: Letters to means have been returned. The data was found not normally distributed hence analysis using ANOVA was not possible. The percentage means and their corresponding standard errors were generated using a Glm model (survival) and a non-parametric analysis (Tukey) was used to compare the means.
Table 4: It seems that no statistical comparison between means in table 4 has made as well. You should perform statistical comparison of means by using a parametric (ANOVA followed by a post hoc test, e.g. SNK) or non-parametric (e.g. Kruskal Wallis) test. Any significant differences should be addressed by letters in the means. Please clarify what statistics you
Response Table 4: The data was subjected to ANOVA analysis followed by a post hoc test Tukey. Significant differences are marked by letters
